# Development of a probabilistic ocean modelling system based on NEMO 3.5: application at eddying resolution

Laurent Bessières[1], Stéphanie Leroux[2], Jean-Michel Brankart[2], Jean-Marc Molines[2],
Marie-Pierre Moine[1], Pierre-Antoine Bouttier[2], Thierry Penduff[2], Laurent Terray[1], Bernard Barnier[2],
and Guillaume Sérazin[1,2]

[1]CNRS/CERFACS, CECI UMR 5318, Toulouse, France.
[2]Univ. Grenoble Alpes, CNRS, IRD, IGE, F-38000 Grenoble, France.

*Correspondence to:* Laurent Bessières (bessieres@cerfacs.fr)

**Abstract.** This paper presents the technical implementation of a new, probabilistic version of the NEMO ocean/sea-ice mod-
elling system. Ensemble simulations with $N$ members running simultaneously within a single executable, and interacting
mutually if needed, are made possible through an enhanced MPI strategy including a double parallelization in the spatial and
ensemble dimensions. An example application is then given to illustrate the implementation, performances and potential use
of this novel probabilistic modelling tool. A large ensemble of 50 global ocean/sea-ice hindcasts has been performed over the
period 1960-2015 at eddy-permitting resolution (1/4°) for the OCCIPUT project. This application is aimed to simultaneously
simulate the intrinsic/chaotic and the atmospherically-forced contributions to the ocean variability, from meso-scale turbulence
to interannual-to-multidecadal time scales. Such an ensemble indeed provides a unique way to disentangle and study both
contributions, as the forced variability may be estimated through the ensemble mean, and the intrinsic chaotic variability may
be estimated through the ensemble spread.

## 1  Introduction

Probabilistic approaches, based on large ensemble simulations, have been helpful in many branches of Earth-system modelling
sciences to tackle the difficulties inherent to the complex and chaotic nature of the dynamical systems at play. In oceanogra-
phy, ensemble simulations have first been introduced for data assimilation purposes, in order to explicitly simulate and, given
observational data, reduce the uncertainties associated to e.g. model dynamics, numerical formulation, initial states, atmo-
spheric forcing (e.g. *Evensen*, 1994; *Lermusiaux*, 2006). This type of probabilistic approach is also used to accurately assess
ocean model simulations against observations (e.g. *Candille and Talagrand*, 2005), or to anticipate on the design of satellite
observational missions (e.g. *Ubelmann et al.*, 2009).
Performing ensemble simulations can be seen as a natural way to take into account the internal variability inherent to any
chaotic and turbulent system, by sampling a range of possible trajectories of this system (independent and identically dis-
tributed). For example, long-term climate projections, or short-term weather forecasts, rely on large ensembles of atmosphere-
ice-ocean coupled model simulations to simulate the probabilistic response of the climate system to various external forcing
scenarii, or to perturbed initial conditions, respectively (e.g. *Palmer*, 2006; *Kay et al.*, 2015; *Deser et al.*, 2016).
The ocean is, like the atmosphere or the full climate system, a chaotic system governed by non-linear equations which couple
various spatio-temporal scales. A consequence is that, in the turbulent regime (i.e. for $1/4^\circ$ or finer resolution), ocean models
spontaneously generate a chaotic intrinsic variability under purely climatological atmospheric forcing, i.e. same repeated an-
nual cycle from year to year. This purely intrinsic variability has a significant imprint on many ocean variables, especially in
eddy-active regions, and develops on spatio-temporal scales ranging from mesoscale eddies up to the size of entire basins, and
from weeks to multiple decades (*Penduff et al.*, 2011; *Grégorio et al.*, 2015; *Sérazin et al.*, 2015). The evolution of this chaotic
ocean variability under repeated climatological atmospheric forcing is sensitive to initial states. This suggests that turbulent
oceanic hindcasts driven by the full range of atmospheric scales (e.g. atmospheric reanalyses) are likely to be sensitive to initial
states as well, and their simulated variability should be interpreted as a combination of the atmospherically-forced and the
intrinsic/chaotic variability.
On the other hand, NEMO climatological simulations at $\sim 2^\circ$ resolution (in the laminar non-eddying regime) driven by a
repeated climatological atmospheric forcing are almost devoid of intrinsic variability (*Penduff et al.*, 2011; *Grégorio et al.*,
2015). Because $\sim 1/4^\circ$-resolution OGCMs are now progressively replacing their laminar counterparts at $\sim$1-2$^\circ$ resolution used
in previous CMIP-type long-term climate projections (e.g. HighResMIP, *Haarsma et al.*, 2016), it becomes crucial to better
understand and characterize the respective features of the intrinsic and atmospherically-driven parts of the ocean variability,
and their potential impact on climate-relevant indices.
Simulating, separating and comparing these two components of the oceanic variability requires an ensemble of turbulent
ocean hindcasts, driven by the same atmospheric forcing, and started from perturbed initial conditions. The high computational
cost of performing such ensembles at global or basin scale explains why only few studies have carried out this type of approach
until now, and with small ensemble sizes (e.g. *Combes and Lorenzo*, 2007; *Hirschi et al.*, 2013).
Building on the results obtained from climatological simulations, the ongoing OCCIPUT project (*Penduff et al.*, 2014) aims
to better characterize the chaotic low-frequency intrinsic variability (LFIV) of the ocean under a fully-varying atmospheric
forcing, from a large (50-member) ensemble of global ocean/sea-ice hindcasts at $1/4^\circ$ resolution over the last 56 years (1960-
2015). The intrinsic and the atmospherically-forced parts of the ocean variability are thus simulated simultaneously under
fully-varying realistic atmosphere, and may be estimated from the ensemble spread and the ensemble mean, respectively. This
strategy also allows to investigate the extent to which the full atmospheric variability may excite, modulate, damp, or pace
intrinsic modes of oceanic variability that were identified from climatological simulations. OCCIPUT mainly focuses on the
interannual-to-decadal variability of ocean quantities having a potential impact on the climate system, such as Sea Surface
Temperature (SST), Meridionnal Overturning Circulation (MOC), and upper Ocean Heat Content (OHC).
This paper presents the technical implementation of the new, fully probabilistic version of the NEMO modelling system
required for this project. It stands at the interface between scientific purposes and new technical developments implemented
in the model. The OCCIPUT project is presented here as an application, to illustrate the system requirements and numerical
performances. The mathematical background supporting our probabilistic approach is detailed in section 2. Section 3 describes
the new technical developments introduced in NEMO to simultaneously run multiple members from a single executable (allow-
ing the online computation of ensemble statistics), with a flexible input/output strategy. Section 4 presents the implementation
of this probabilistic model to perform regional and global 1/4° ensembles, both performed in the context of OCCIPUT. The
strategy chosen to trigger the growth of the ensemble spread, and the numerical performances of both implementations are
also discussed. Section 5 finally presents some preliminary results from OCCIPUT to further illustrate potential scientific
applications of this probabilistic approach. A summary and some concluding remarks are given in section 6.

## 6 2 From deterministic to probabilistic ocean modelling: mathematical background

The classical, deterministic ocean model formulation can be written as
$$d\mathbf{x} = \mathcal{M}(\mathbf{x}, t)dt \tag{1}$$
where $\mathbf{x} = (x_1, x_2, ..., x_N)$ is the model state vector; $t$ is time; and $\mathcal{M}$ is the model operator, containing the expression of the
tendency for every model state variable. An explicit time-dependence is included in the model operator since the tendencies
depend on the time-varying atmospheric forcing.
Computing a solution to Eq. (1) requires the specification of the initial condition at $t = 0$, from which the future evolution
of the system is fully determined. OCCIPUT investigates how perturbations in initial conditions evolve and finally affect the
statistics of climate-relevant quantities. This problem may be addressed probabilistically by solving the Liouville equation:
$$\frac{\partial p(\mathbf{x}, t)}{\partial t} = -\sum_{k=1}^{N} \frac{\partial}{\partial x_k} \left[ \mathcal{M}(\mathbf{x}, t) p(\mathbf{x}, t) \right] \tag{2}$$
where $p(\mathbf{x}, t)$ is the probability distribution of the system state at time $t$. Eq. (2) shows that this distribution is simply advected in
the phase space by local model tendencies. In chaotic systems, small uncertainties in the initial condition ($p(\mathbf{x}, 0)$ concentrated
in a very small region of the phase space) yield diverging trajectories; such systems are poorly predictable on the long range.
In the turbulent ocean, the mesoscale turbulence and the low-frequency intrinsic variability (LFIV) have a chaotic behaviour.
They both belong to what we will call more generally intrinsic variability in sections 4 and 5, in the sense that they do not
result from the forcing variability but spontaneously emerge from the ocean even with constant or seasonal forcing. Because
this intrinsic variability is chaotic, it can only be described in a statistical sense and the probabilistic approach of Eq. (2) is
required.
In addition to uncertainties in the initial condition, it is sometimes useful to assume that the model dynamics itself is un-
certain. This leads to a non-deterministic ocean model formulation, in which model uncertainties are decribed by stochastic
processes. One possibility is for instance to modify Eq. (1) as follows:
$$d\mathbf{x} = \mathcal{M}(\mathbf{x}, t)dt + \Sigma(\mathbf{x}, t)d\mathbf{W}_t \tag{3}$$
In this equation, $\mathbf{W}_t$ is an M-dimensional standard Wiener process, and $\Sigma(\mathbf{x}, t)$ is an $N \times M$ matrix, describing the influence of
these processes on the model tendencies. Eq. (3) does not include all possible ways of introducing a stochastic parameterization
in a dynamical model, but it is sufficient to include the implementation that is described in this paper (in particular, to include

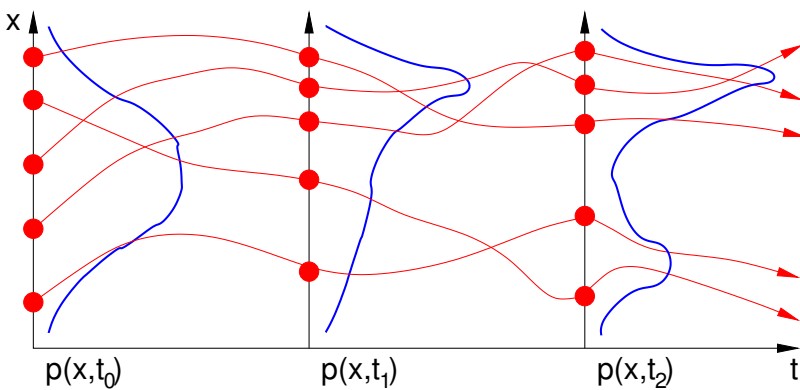

**Figure 1.** Schematic of an ensemble simulation (red trajectories), as an approximation to the simulation of an evolving probability distribution (in blue).

the use of space-correlated or time-correlated autoregressive processes by expanding the definition of $\mathbf{x}$ in Eq. (3)). With this additional stochastic term, Liouville equation transforms to Fokker-Planck equation:

$$
\frac{\partial p(\mathbf{x},t)}{\partial t} = -\sum_{k=1}^{N} \frac{\partial}{\partial x_k}\left[\mathcal{M}(\mathbf{x},t)p(\mathbf{x},t)\right]
$$

$$
+ \frac{1}{2}\sum_{k=1}^{N}\sum_{l=1}^{N}\frac{\partial^2}{\partial x_k \partial x_l}\left[D_{kl}(\mathbf{x},t)p(\mathbf{x},t)\right] \tag{4}
$$

where $D_{kl}(\mathbf{x},t) = \sum_{p=1}^{M}\Sigma_{jp}(\mathbf{x},t)\Sigma_{lp}(\mathbf{x},t)$. The probability distribution $p(\mathbf{x},t)$ is thus affected by the stochastic diffusion tensor $\mathbf{D}(\mathbf{x},t)$ during its advection in the phase space by local model tendencies.

However, since Eqs. (2) and (4) are partial differential equations in an N-dimensional space, they generally cannot be solved explicitly for large size systems. Only an approximate description of $p(\mathbf{x},t)$ can be obtained in most practical situations. A common solution is to reduce the description of $p(\mathbf{x},t)$ to a moderate size sample, which can be viewed as a Monte Carlo approximation to Eqs. (2) and (4). This approach is illustrated in Figure 1. The computation is initialized by a sample of the initial probability distribution $p(\mathbf{x},t_0)$ (on the left in the figure), and each member of the sample is used as a different initial condition to Eqs. (1) and (3). The classical model operator can then be used to produce an ensemble of model simulations (red trajectories in the figures), which provide a sample of the probability distribution at any future time, e.g. $p(\mathbf{x},t_1)$, or $p(\mathbf{x},t_2)$. This Monte Carlo approach is very general and can be also applied to any kind of stochastic parameterization (not only the particular case described by Eq. (3)). It was first applied to ocean models in the framework of the ensemble Kalman filter (*Evensen*, 1994) to solve ocean data assimilation problems.

In summary, Eq. (1) describes the problem that is classically solved by the NEMO model; Eq. (3) is a modification of this problem with stochastic perturbations of the model equations that explicitly simulate model uncertainties; in this paper, this problem is solved using an ensemble simulation, which provides identically-distributed realizations from the probability distribution, and thus a way to compute any statistic of interest.

## 3  Performing ensemble simulations with NEMO

The NEMO model (Nucleus for a European Model of the Ocean), described in *Madec* (2012), is used for oceanographic research, operational oceanography, seasonal forecasts and climate studies. This system embeds various model components (see http://www.nemo-ocean.eu/), including a circulation model (OPA, Océan PArallélisé), a sea-ice model (LIM, Louvain-la-Neuve Ice model), and ecosystem models with various levels of complexity. Every NEMO component solves partial differential equations discretized on a three-dimensional grid using finite-difference approximations. The purpose of this section is to present the technical developments introduced in our probabilistic NEMO version, and to make the connection between these new developments and existing NEMO features.

### 3.1  Ensemble NEMO parallelization

The standard NEMO code is parallelized with MPI (Message Passing Interface) using a domain decomposition method. The model grid is divided in rectangular subdomains ($i = 1, \ldots, n$), so that the computations associated to each subdomain can be performed by a different processor of the computer. Spatial finite-difference operators require knowledge of the neighbouring grid points, so that the subdomains must overlap to allow the application of these operators on the discretized model field. Whenever needed, the overlapping regions of each subdomain must be updated using the computations made for the neighbouring subdomains. The NEMO code provides standard routines to perform this update. These routines use MPI to get the missing information from the other processors of the computer. This communication between processors makes the connection between subdomains in the model grid.

In practice, upon initialization one MPI communicator is defined with as many processors as subdomains, each processor is associated with a subdomain and knows which are its neighbours.

Ensemble simulations may be performed with NEMO by a direct generalization of the standard parallelization procedure described above. In other words, our ensemble simulations are performed from one single call to the NEMO executable, simply using more processors to run all members in parallel. This technical option is both natural and unnatural. It is natural since an ensemble simulation provides an approximate description of the probability distribution; it is thus conceptually appealing to advance all members together in time. It is unnatural since independent ensemble members may be run separately (in parallel, or successively) using independent calls to NEMO. However, the solution we propose is so straightforward that there is virtually no implementation cost, and is more flexible since the ensemble members may be run independently, by groups of any size, or all together. Furthermore, running all ensemble members together provides a new interesting capability: the characteristics of the probability distribution $p(\mathbf{x}, t)$ in Eq. (2) or (4) may be computed online, virtually at every time step of the ensemble simulation. This has been done using MPI to gather the required information from every member of the ensemble. These MPI communications make a natural connection between ensemble members, as a sample of the probability distribution $p(\mathbf{x}, t)$.

In practice, this implementation option only requires that at the beginning of the NEMO simulation, one MPI communicator is defined for each ensemble member, each one with as many processors as subdomains, so that each processor knows to which member it belongs, on which subdomain it is going to compute and what are its neighbours. Inside each of these

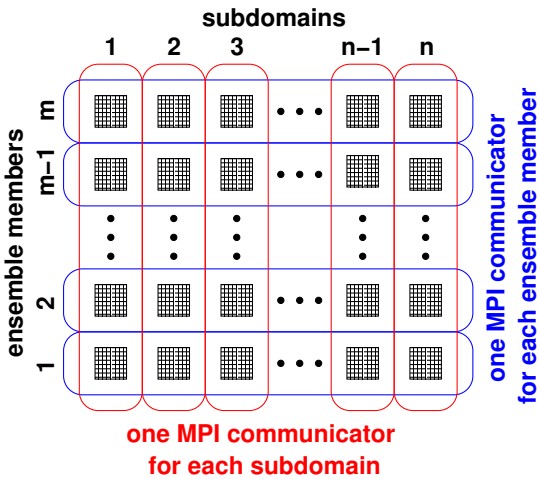

**Figure 2.** Schematic of the double parallelization introduced in NEMO: each processor (black squares) is dedicated to the computations associated to one model subdomain and one ensemble member. There is one MPI communicator within each ensemble member (in blue) to allow communications between neighbouring subdomains as in the standard NEMO parallelization; and there is one MPI communicator within each model subdomain (in red) to allow communication between ensemble members (e.g. to compute ensemble statistics online if needed). The total number of processors is thus equal to the product of the ensemble size by the number of subdomains ($m \times n$).

communicators, each ensemble member may be run independently from the other members, without changing anything else
in the NEMO code. However, all members are obviously not supposed to behave exactly the same: the index of the ensemble
member must have some influence on the simulation. This influence may be in the name of the files defining the initial
condition, parameters or forcing, or in the seeding of the random number generator (if a random forcing is applied, as in Eq. 3).
The index of the ensemble member must also be used to modify the name of all output files, so that the output of different
members is saved in different files. As it appears, this implementation of ensemble simulations does not require much coding
effort (a few tens of lines in NEMO, partly because most of the basic material was already available in the original code). More
technical details about this can be found in appendix 7.
In summary, the NEMO ensemble system relies on a double parallelization, over model subdomains and over ensemble
members, as illustrated in Figure 2. In this algorithm, ensemble simulations are thus intricately linked to MPI parallelization.
There is no explicit loop over the ensemble members; this loop is done implicitly through MPI; running more ensemble
members means either using more processors or using less processors for each member.

## 16 3.2 Online ensemble diagnostics

As mentioned above, one important novelty offered by the ensemble NEMO parallelization is the ability to compute online any
feature of the probability distribution $p(\mathbf{x},t)$. This can be done within additional MPI communicators connecting all ensemble
members for each model subdomain (in red in Fig. 2). MPI sums in these communicators are for instance immediately sufficient
to estimate:

– the mean of the distribution

$$\mu_k = \int x_k p(\mathbf{x}, t) d\mathbf{x} \quad \sim \quad \tilde{\mu}_k = \frac{1}{m} \sum_{j=1}^{m} x_k^{(j)} \tag{5}$$

where $x_k$ is one of the model state variable; $x_k^{(j)}$ is this variable simulated in member $j$; $\mu_k$ the mean of the distributon for this variable; and $\tilde{\mu}_k$, the estimate of the mean obtained from the ensemble. It is interesting to note that the sum over ensemble members in Eq. (5) is not explicitly coded in NEMO, it is performed instead by a single call to MPI, which computes the sums over all processors of the ensemble communicators (in red in Fig. 2). The same remark also applies to the sums in the following equations.

– the variance of the distribution

$$\sigma_k^2 = \int (x_k - \mu_k)^2 p(\mathbf{x}, t) d\mathbf{x}$$

$$\sim \quad \tilde{\sigma}_k^2 = \frac{1}{m-1} \sum_{j=1}^{m} \left( x_k^{(j)} - \tilde{\mu}_k \right)^2$$

where $\sigma_k^2$ is the variance of the distributon for variable $x_k$; and $\tilde{\sigma}_k^2$, the estimate obtained from the ensemble. The ensemble standard deviation is simply the square root of $\tilde{\sigma}_k^2$.

– Ensemble covariance between 2 variables at the same model grid point:

$$\gamma_{kl} = \int (x_k - \mu_k)(x_l - \mu_l) p(\mathbf{x}, t) d\mathbf{x}$$

$$\sim \quad \tilde{\gamma}_{kl} = \frac{1}{m-1} \sum_{j=1}^{m} \left( x_k^{(j)} - \tilde{\mu}_k \right) \left( x_l^{(j)} - \tilde{\mu}_l \right)$$

where $\gamma_{kl}$ is the covariance between variables $x_k$ and $x_l$, and $\tilde{\gamma}_{kl}$, the estimate obtained from the ensemble.

This is directly generalizable to the computation of higher order moments (skewness, kurtosis), which is then reduced to MPI sums in the ensemble communicators. Moreover, simple MPI algorithms can also be designed to compute online many other probabilistic diagnostics, as for instance the rank of each member in the ensemble, and from there, estimates of quantiles of the probability distribution. Specific applications of this feature are discussed in section 5.

This online estimation of the probability distribution, via the computation of ensemble statistics, opens another interesting new capability: the solution of the model equations may now depend on ensemble statistics, available at each time step if needed. For instance, it may be interesting to relax the modeled forced variability towards reference (e.g. reanalyzed or climatological) fields, with no explicit damping of the intrinsic variability: the nudging term would involve the current ensemble mean and be applied identically to all members at the next time step, resulting in a simple "translation" of the entire ensemble distribution toward the reference field.

Other applications, such as ensemble data assimilation, may also require an online control of the ensemble spread, which is hereby made possible within NEMO.

### 3.3 Connection with NEMO stochastic parameterizations

Ensemble simulations are directly connected to stochastic parameterizations (as introduced in Eq. 3). In NEMO, stochastic parameterizations have recently been implemented to explicitly simulate the effect of uncertainties in the model (*Brankart et al.*, 2015). In practice, this is done by generating maps of autoregressive processes, which can be used to introduce perturbations in any component of the model. In *Brankart et al.* (2015), examples are provided to illustrate the effect of these perturbations in the circulation model, in the ecosystem model and in the sea ice model. For instance, a stochastic parameterization was introduced in the circulation model to simulate the effect of unresolved scales in the computation of the large scale density gradient, as a result of the nonlinearity of the sea water equation of state (*Brankart*, 2013). This particular stochastic parameterization is switched on during one year in order to initiate the dispersion of the OCCIPUT ensemble simulations started from a single initial condition (see section 4).

### 3.4 Connection with NEMO data assimilation systems

Ensemble model simulations are also key in ensemble data assimilation systems: they propagate in time uncertainties in the model initial condition, and provide a description of model uncertainties in the assimilation system (e.g. using stochastic perturbations). Data assimilation can then be carried out by conditioning this probability distribution to the observations whenever they are available. The ensemble data assimilation method that is currently most commonly used in ocean applications is the Ensemble Kalman filter (*Evensen*, 1994), which performs the observational update of the model probability distributions with the assumption that they are Gaussian. However, it has been recently suggested that the Gaussian assumption is often insufficient to correctly describe ocean probability distributions, and that more general methods using for instance anamorphosis transformations (*Bertino et al.*, 2003; *Brankart et al.*, 2012) or a particle filtering approach (e.g. *Van Leeuwen*, 2009) may be needed. One of the purpose of the SANGOMA European project is precisely to develop such more general methods for ocean applications, and to implement them within NEMO-based ocean data assimilation systems (e.g. *Candille et al.*, 2015). In these methods, the role of ensemble NEMO simulations is even more important since they require a more detailed decription of the probability distributions (as compared to the Gaussian assumption, which only requires the mean and the covariance). The importance of ensemble simulations in data assimilation certainly explains why the ensemble NEMO parallelisation (introduced above) has been first applied within SANGOMA, to assimilate altimetric observations in an eddying NEMO configuration of the North Atlantic (*Candille et al.*, 2015).

### 3.5 Connection with the NEMO observation operator and model assessment metrics

Another important benefit of the probabilistic approach is to consolidate and objectivate statistical comparisons between actual observations and model-derived ensemble synthetic observations. Probabilistic assessment metrics are commonly used in the atmospheric community (e.g. *Toth et al.*, 2003) but are quite new in oceanography. Briefly speaking, these methods generally quantify two attributes of an ensemble simulation: the *reliability* and the *resolution*. An ensemble is reliable if the simulated probabilities are statistically consistent with the observed frequencies. The ensemble resolution is related to the system ability to

discriminate between distinct observed situations. If the ensemble is reliable, the resolution is directly related to the information content (or the spread) of the probability distribution. A popular measure of these two attributes is for instance provided by the Continuous Rank Probability Score (CRPS), which is based on the square difference between a cumulative distribution function (cdf) as provided by the ensemble simulation and the corresponding cdf of the observations (*Candille and Talagrand*, 2005).

In OCCIPUT, such probabilistic scores will be computed from real observations and from the ensemble synthetic observations (along-track Jason-2 altimeter data and ENACT-ENSEMBLE temperature and salinity profile data) generated online using the existing NEMO observation operator (NEMO-OBS module). NEMO-OBS is used exactly as in standard NEMO within each member of the ensemble, hence providing an ensemble of model equivalents for each observation rather than a single value. Probabilistic metrics (i.e. CRPS score) will then be computed to assess the reliability and resolution of the OCCIPUT simulations.

## 3.6 Connection with NEMO I/O strategy

Our implementation of ensemble NEMO using enhanced parallelization is technically not independent from the NEMO I/O strategy. In NEMO indeed, the input and output of data is managed by an external server (XIOS, for XML IO Server), which is run on a set of additional processors (not used by NEMO). The behavior of this server is controlled by an XML file, which governs the interaction between XIOS and NEMO, and which defines the characteristics of input and output data: model fields, domains, grid, I/O frequencies, time averaging for outputs,... To exchange data with disk files, every NEMO processor makes a request to the XIOS servers, consistently with the definitions included in the XML file. In this operation, the XIOS servers buffer data in memory, with the decisive advantage of not interrupting NEMO computations with the reading or writing in disk files. One peculiarity of this buffering is that each XIOS server reads and writes one stripe of the global model domain (along the second model dimension), and thus exchanges data with processors corresponding to several model subdomains. To optimize the system, it is obviously important that the number of XIOS servers (and thus the size of these stripes) be correctly dimensioned according to the amount of I/O data, which may heavily depend on the model configuration and on the definition of the model outputs.

To use XIOS with our implementation of ensemble NEMO for OCCIPUT, we thus had to take care of the two following issues. First, different ensemble members must write different files. This problem could be solved because XIOS was already designed to work with a coupled model, and can thus deal with multiple contexts, one for each of the coupled model components. It was thus directly possible to define one context for each ensemble member, just as if they were different components of a coupled model. Second, in ensemble simulations, the amount of output data is proportional to the ensemble size, so that the number of XIOS servers must be increased accordingly, with some care however, because the size of the data stripe that is processed by each server should not be reduced too much.

# 4 Example of application: the OCCIPUT project

The implementation of this ensemble configuration of NEMO was motivated to a large extent by the scientific objectives of the OCCIPUT project, described in the introduction. In this section, we present two ensemble simulations, E-NATL025 and E-ORCA025, performed in the context of this project. We focus on the model set-up, the integration strategy, the numerical performances of the system, followed by a few illustrative preliminary results in section 5.

## 4.1 Regional and global configurations

E-ORCA025 is the main ensemble simulation aimed by OCCIPUT. It is a 50-member ensemble of global ocean/sea-ice hindcasts at 1/4° horizontal resolution, run for 56 years. Before performing this large ensemble, a smaller (20-year × 10-member) regional ensemble simulation, E-NATL025, was performed on the North Atlantic domain in order to test the new system implementation and to validate the stochastic perturbation strategy for triggering the growth of the ensemble dispersion. The global and the regional ensemble configurations are both based on version 3.5 of NEMO, and use a 1/4° eddy-permitting quasi-isotropic horizontal grid (∼27 km at the equator), the grid size decreasing poleward. Table 1 summarizes the characteristics of ensembles. The model parameters are very close to those used in the DRAKKAR-ORCA025[1] one-member setups (*Barnier et al.*, 2006), the present setup using a greater number (75) of vertical levels (see table 1). They are also close to those used for the 327-year ORCA025-MJM01 one-member climatological simulation used in *Penduff et al.* (2011), *Grégorio et al.* (2015) and *Sérazin et al.* (2015) to study various imprints of the LFIV under seasonal atmospheric forcing.

## 4.2 Integration and stochastic perturbation stategies

A one-member spin-up simulation is first performed for each ensemble. For the regional ensemble (E-NATL025), it is performed from 1973 (cold start) to 1992, forced with DFS.5.2 atmospheric conditions (*Dussin et al.*, 2016). For the global ensemble (E-ORCA025), the spin-up strategy has to be adapted to match the OCCIPUT objective to perform the ensemble hindcast over the longest period available in the atmospheric forcing DFS5.2 (i.e. 1960-2015). The one-member spin-up simulation is thus performed as follows: (1) it is first forced by the standard DFS5.2 atmospheric forcing from January 1st, 1958 (cold start) to December 31st, 1976; (2) this simulation is continued over January 1977 with a modified forcing function that linearly interpolates between the 1st of January 1977 to the 31st of January 1958; (3) the standard DFS5.2 forcing is applied again normally from February 1st, 1958 to the end of 1959. This 21-year spin-up (1958 to 1977, then 1958 to 1959) thus includes a smooth artificial transition from January 1977 back to January 1958. This choice was made as a compromise to maximize the duration of the single-member spin-up simulation and of the subsequent ensemble hindcast, while minimizing the perturbation in the forcing during the transition, since 1977 was found to be a reasonable analogue of 1958 in terms of key climate indices (El Niño Southern Oscillation, North Atlantic Oscillation, and Southern Annular Mode).

The N members of both ensemble simulations (i.e. N=10 for E-NATL025 and N=50 for E-ORCA025) are started at the end of the single-member spin-up; a weak stochastic perturbation in the density equation, as described by equation 3 and in section

---

[1]DRAKKAR-ORCA025 website: http://www.drakkar-ocean.eu/global-models/orca025

|  | REGIONAL ENSEMBLE | GLOBAL ENSEMBLE |
|---|---|---|
| ENSEMBLE NAME: | E-NATL025 | E-ORCA025 |
| SPATIAL DOMAIN: | North Atlantic (21°S-81°N) | Global |
| HORIZONTAL RESOLUTION: | 1/4° (486×530 grid-points) | 1/4° (1441×1021 grid-points) |
| VERTICAL RESOLUTION: | 46 levels | 75 levels |
| ENSEMBLE SIZE: | 10 members | 50 members |
| TIME PERIODE COVERED BY ENSEMBLE: | 1993-2012 | 1960-2015 |
| STOCHASTIC PERTURBATION PHASE: | 1 year (1993) | 1 year (1960) |
| SURFACE BOUNDARY CONDITIONS: | DFS5.2 (*Dussin et al.*, 2016) Turbulent air-sea fluxes : bulk formula. (with absolute wind) SSS relaxation in each member: 50m/(300 days) piston velocity. | |

**Table 1.** Main characteristics of the NEMO 3.5 set-up used for the regional and global OCCIPUT ensembles.

3.3 (see also *Brankart et al.*, 2015) is then activated within each member. This stochastic perturbation is only applied for one year to seed the ensemble dispersion (during 1993 for E-NATL025, during 1960 years for E- ORCA025). It is then switched off throughout the rest of the ensemble simulations. Once the stochastic perturbation is stopped, the N members are thus integrated from slightly perturbed initial conditions (i.e. 19 more years for E-NATL025 and 55 more years for E-ORCA025), but forced by the exact same atmospheric conditions (DFS5.2, *Dussin et al.*, 2016). The code is parallelized with the double-parallelization technique described in 3.1 so that the N members are integrated simultaneously through one single executable.

**4.3   Performance of the NEMO ensemble system in OCCIPUT configurations**

The regional ensemble (E-NATL025) was performed to test the system implementation and to calibrate the global configuration. The global ensemble simulation E-ORCA025 represents in total 2821 cumulated years of simulation (56 yrs × 50 members + 21 yrs of one-member spin-up ) over 110 million grid points (Lon × Lat × Depth = 1442×1021×75). As confirmed thereafter in Figure 3, integrating such a system within one executable with reasonable wall-clock time and managing its outputs lies beyond national or regional European centres computational capabilities (i.e Tier-1 systems) and requires systems that can provide European top capabilities, which are beyond the Petaflops level (i.e Tier-0 systems).

All simulations were performed between 2014 and 2016 on the French Tier-0 Curie supercomputer, supported by PRACE (Partnership for Advanced Computing in Europe) and GENCI (Grand Equipement National de Calcul Intensif, French rep-

resentative in PRACE) grants ($19.10^6$ HCPU, see details below). Curie is a Bull system (BullX series designed for extreme computing) based on Intel processors. The architecture used for the simulations is the one of the "Curie thin nodes" configuration (Curie-TN), which is mainly targeted at MPI parallel codes and includes more than $80,000$ Intel's Sandy-Bridge computing cores (Peak frequency per core: 2.7 GHz) gathered in 16-cores nodes of 64 GB of memory.

Preliminary tests showed that the one-member ORCA025 configuration has a good scalability up to 400 cores on Curie-TN (not shown). In order to test the ensemble global configuration on Curie-TN, short 180-step experiments were run, disregarding the first and last steps (which correspond to reading and writing steps, respectively, that are performed only once during production jobs). The performance of the system was measured in steps per minute by analyzing the 160 steps in between (steps 10 to 170). Figure 3.a shows this measure of the system performance (in step/min) as a function of the number of members, for different domain decompositions (64, 128, 160, 256 and 400 cores/member). It appears that the performance is independent of the ensemble size for domain decomposition up to 160 domains per member. When more than 160 domains per member are used, the performance starts to decrease for increasing ensemble size, from 25 members (resp. 10) for the decomposition with 256 (resp. 400) domains per member. Fluctuations in step per minute may appear (see the performance for the decomposition with 400 domains per member and 25 members on figure 3.a), depending on machine load and files system stability (the performance of this specific point has not been reassessed for CPU cost reasons). The scalability of the global ensemble configuration E-ORCA025 as aimed in OCCIPUT ($N$=50) is shown in Figure 3.b: the efficiency is measured as the ratio of the observed speedup to the theoretical speedup, relative to the smallest domain decomposition tested, i.e. with 3200 cores ($50 \times 64$). The efficiency is remarkably good and remains around 90% for 20.000 used cores.

Based on these performance tests, a domain decomposition with relatively few cores was chosen in order to maintain a manageable rate of I/Os. The decomposition with 128 cores per member has been retained (corresponding to the red line on Figure 3) so that 50x128=6400 cores are used for the ensemble-NEMO system.

In order to optimize and to make the I/O data flux management flexible, 40 XIOS servers have been run as independant MPI tasks in detached mode, allowing the overlap of I/O operations with computations. Compared to the 10-member regional case, the 50-member global case required a larger XIOS buffer size. For this reason, each of the 40 XIOS instances was run on a dedicated and exclusive Curie "thin node", allowing each server to use the entire memory available on each 16-core node (i.e 64 GB); the 40 XIOS servers thus used $16\times40$=640 cores in total. The integration of the 50-member global E-ORCA025 ensemble therefore required the use of 6400+($40\times16$) = 7040 cores.

XIOS makes use of parallel file systems capabilities via the Netcdf4-HDF5 format, that allows both online data compression and parallel I/O. Therefore, XIOS is used in "multiple file" mode where each XIOS instance writes a file for one stripe of the global domain, yielding 40 files times 50 members for each variable and each time. At the end of each job, the 40 stripes are recombined on-the-fly into global files.

Preliminary tests have shown that the 50-member E-ORCA025 global configuration performs about 20 steps per minute, including actual I/O fluxes and additional operations (e.g. production of ensemble synthetic observations). Since the numerical stability of this global setup requires a model time step of 720s, about 2 million time steps, 85 days of elapse time, and about 14.4 million core hours were needed in theory to perform the 56-year OCCIPUT ensemble simulation. The final CPU

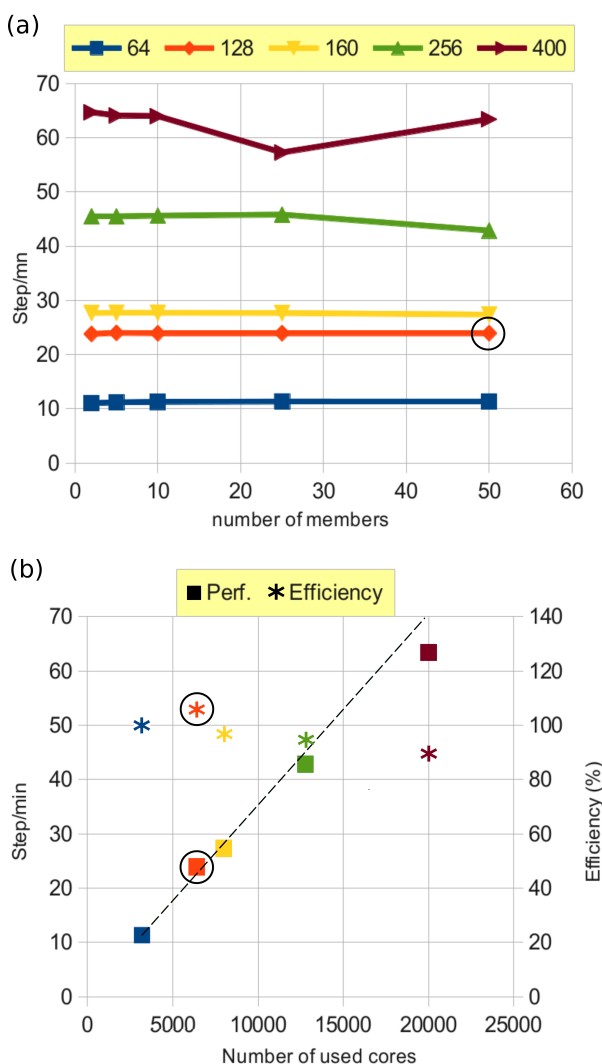

**Figure 3.** (a) Performance of the global ensemble configuration as a function of ensemble size $N$, for five domain decompositions: 64, 128, 160, 256 and 400 cores per member (colored lines). (b) Performance in steps per minute and efficiency in % of the global ensemble configuration with 50 members. The dotted line represents the theorical speedup. The number of cores corresponds here to $N$ times the number of subdomains per member. Our final choice (50 members, 128 cores per member) is indicated with black circles.

cost of the global ensemble experiment was about 19 million CPU hours, due to fluctuations in model efficiency, occasional
problems on file systems which required the repetition of certain jobs, the need to decrease the model time step (increased
high-frequency variance in the wind forcing data over the last decades) and the online computation of ensemble diagnostics
(high-frequency ensemble covariances, all terms of the heat content budget ensemble) over the last decade. The cost of online
ensemble diagnostics depends on the call frequency, number and size of the concerned fields, on the architecture of the machine
and the performance of communications. Our online ensemble diagnostics concerned a few two-dimensional fields at hourly
to monthly frequencies, and had a negligeable cost.
The final E-ORCA025 global database is saved in Netcdf4-HDF5 format (chunked and compressed, compression ratio in
*italics* below). The primary dataset produced by the model consists in the following: monthly averages for full-3D fields (56 yr
$\times$ 12 months $\times$ 50 members $\times$ 2.8 GB x *41.5%* = 39 TB), 5-day averages for sixteen 2D-fields (56 yr $\times$ 50 members $\times$ 6.8 GB
$\times$ *30%* = 6TB), the Jason-2 and ENACT-ENSEMBLES ensemble synthetic observations (5TB), and hourly ensemble statistics
for key variables (1 TB). One restart file per member and per year is also archived (about 35 TB after compression). We then
computed a secondary dataset, consisting in 50-member yearly/decadal averages of the 3D-fields (2 TB), ensemble deciles of
monthly/yearly/decadal 3D-fields (6 TB), and data associated with on-line monitoring (1 TB). The total output amounts to less
than 100TB and 100.000 inodes on the Curie-TN file system.

## 20   5   Preliminary results from the OCCIPUT application

We now present some preliminary results from the regional and global OCCIPUT ensemble simulations described in section
4.1, in order to illustrate the concepts and the technical implementation presented above.

### 23   5.1   Probabilistic interpretation

Figure 4 shows for the 10-member regional ensemble the 1993-2012 timeseries of monthly temperature anomalies at depth 93
m at two contrasting grid points: in the Gulf Stream and in the middle of the North Atlantic subtropical gyre. Panels a. and c.
represent a sample of $N$ trajectories of the temperature given the identical atmospheric evolution that forces all members.
These temperature anomalies were computed by first removing the long-term non-linear trend of the timeseries derived from
a local regression model (as in *Grégorio et al.*, 2015). This detrending step acts as a non-linear high-pass temporal filter with
negligible end-point effect (LOESS detrending, e.g. *Cleveland et al.*, 1992; *Cleveland and Loader*, 1996), which successfully
removes the unresolved imprints of very low-frequency variabilities (of forced or intrinsic origin), and possible non-linear
model drifts. We focus here on the ocean variability that is fully resolved in the 20-yr regional simulation output; we thus
choose to remove the total long-term trend of each member individually prior to plotting/analyzing the ensemble statistics
presented here. The mean seasonal cycle computed over the ensemble has also been removed from the monthly timeseries.

3       The ensemble-mean timeseries (thereafter E-mean, also noted $\tilde{\mu}_k$ in section 3) was then computed from these detrended

timeseries, and illustrates the temperature evolution common to all members, i.e. forced by the atmospheric variability. The

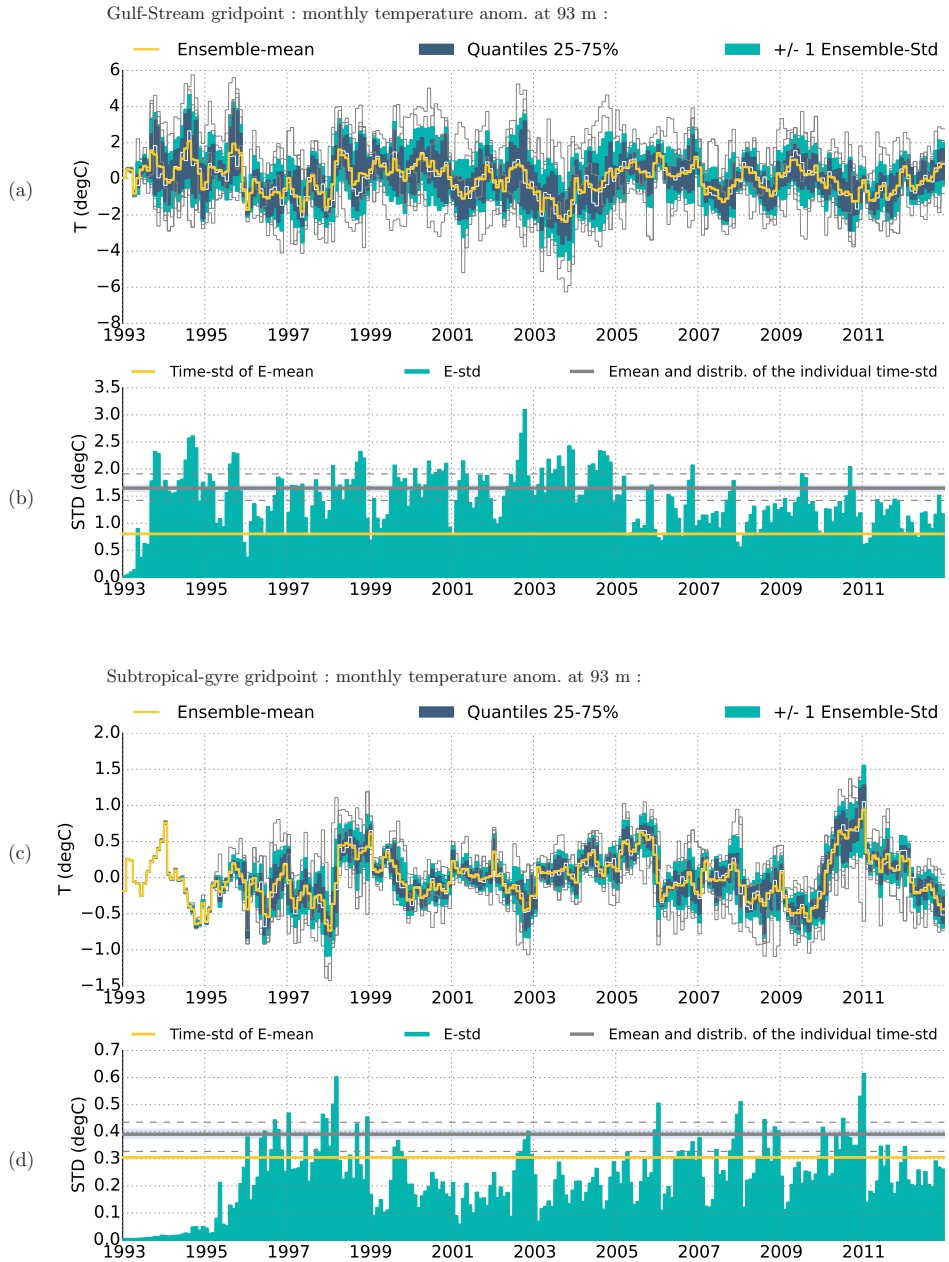

**Figure 4.** Ensemble statistics of the monthly temperature anomalies from the regional ensemble E-NATL025, at depth 93 m at two grid-points: (a,b) in the Gulf-Stream (42°N;56°W), and (c,d) in the North-Atlantic subtropical gyre (22°N;42°W). Anomalies are shown after detrending and seasonal cycle removed (see text for details). (a) The individual trajectories with time of the 10 members appear in thin grey. E-mean is in thick yellow, the interval between quantiles Q1(25%) and Q3 (75%) is filled in dark blue, and the interval E-mean +/- one E-STD is filled in green. (b) E-STD (intrinsic variability, green shading) is compared to the Time-STD of E-mean (forced variability, thick yellow line). Also shown in (b) is the distribution of Time-STD for the 10 members: ensemble mean of the Time-STDs (solid grey), minimum and maximum (dashed grey), and mean +/- one ensemble standard deviation (pale blue shading).

temporal standard deviation (thereafter Time-STD) of this ensemble mean thus provides an estimate of the atmospherically-
forced variability.
The dispersion of individual timeseries about the ensemble mean indicates the amount of intrinsic chaotic variability gener-
ated by the model. Its time-varying magnitude may be estimated by the ensemble standard deviation (thereafter E-STD, also
noted $\tilde{\sigma}_k$ in section 3). Besides these low-order statistical moments, ensemble simulations actually provide an estimate of the
full ensemble probability density function distribution (E-PDF) at any time, with an accuracy that increases with the number
of members in the ensemble (see also section 3.2).

## 5.2    Initialization and evolution of the ensemble spread

Unlike in short-range ensemble forecast exercises, we do not seek here to maximize the growth rate of the initial dispersion;
we let the model feed the spread and control its evolution following its physical laws. Panels b. and d. in Figure 4 confirm
that the stochastic perturbation strategy (section 4.2) successfully seeds an initial spread between the ensemble members. The
evolution and growth rate of the temperature E-STD depend on the geographical location: it grows faster in turbulent areas
such as the Gulf Stream (Fig. 4.b) and slower in less-turbulent areas like the subtropical gyre (Figure 4.d). Note that the
spread keeps growing after the stochastic parametrization has been switched off at the end of 1993, and tends to reach some
leveled/saturated value after a few years. It is nevertheless still subject to clear modulations of its magnitude on time-scales
ranging from monthly to interannual. An additional 8-year experiment (not shown here) has confirmed that when the small
stochastic perturbation is applied over the whole simulation instead of one year, the overall evolution, magnitude, and spatial
patterns of E-STD, and the ensemble mean solution remain unchanged. In other words, the stochastic parametrization seeds
the spread during the initialization period, but the subsequent evolution and magnitude of intrinsic variability is subsequently
controlled by the model non-linearities regardless of the initial stochastic seeding.

## 5.3    Spatial patterns of the ensemble spread

Figure 5 shows maps of E-STD in the regional ensemble E-NATL025, computed from annual-mean anomalies of sea surface
height (SSH), sea surface temperature (SST), and temperature at 93 m and 227 m over the last simulated year (i.e. 2012). These
maps thus quantify the imprint of interannual intrinsic variability on these variables, and show that after 20 years of simulation,
the ensemble spread has cascaded from short (mesoscale-like) periods to long timescales. Annual E-STDs reach their maxima
in eddy-active regions like the Gulf Stream (Fig. 5.a) and the North Equatorial Counter Current (Fig. 5.c) where hydrodynamic
instabilities are strongest and continuously feed mesoscale activity (i.e. small-scale intrinsic variability), which then cascades
to longer time scales. The order of magnitude of this low-frequency intrinsic variability (LFIV) is about 1 °C for SST and 10
cm for SSH in the Gulf Stream in 2012. We will compare these amplitudes to those of the atmospherically-forced variability
(Time-STD of E-mean) in the next section.
Comparing panels b., c. and d. in Figure 5 also illustrates that the ensemble spread of yearly temperature (i.e. its low-
frequency intrinsic variability) peaks at subsurface (around the thermocline), and tends to decrease toward the surface in
eddy-quiet regions.

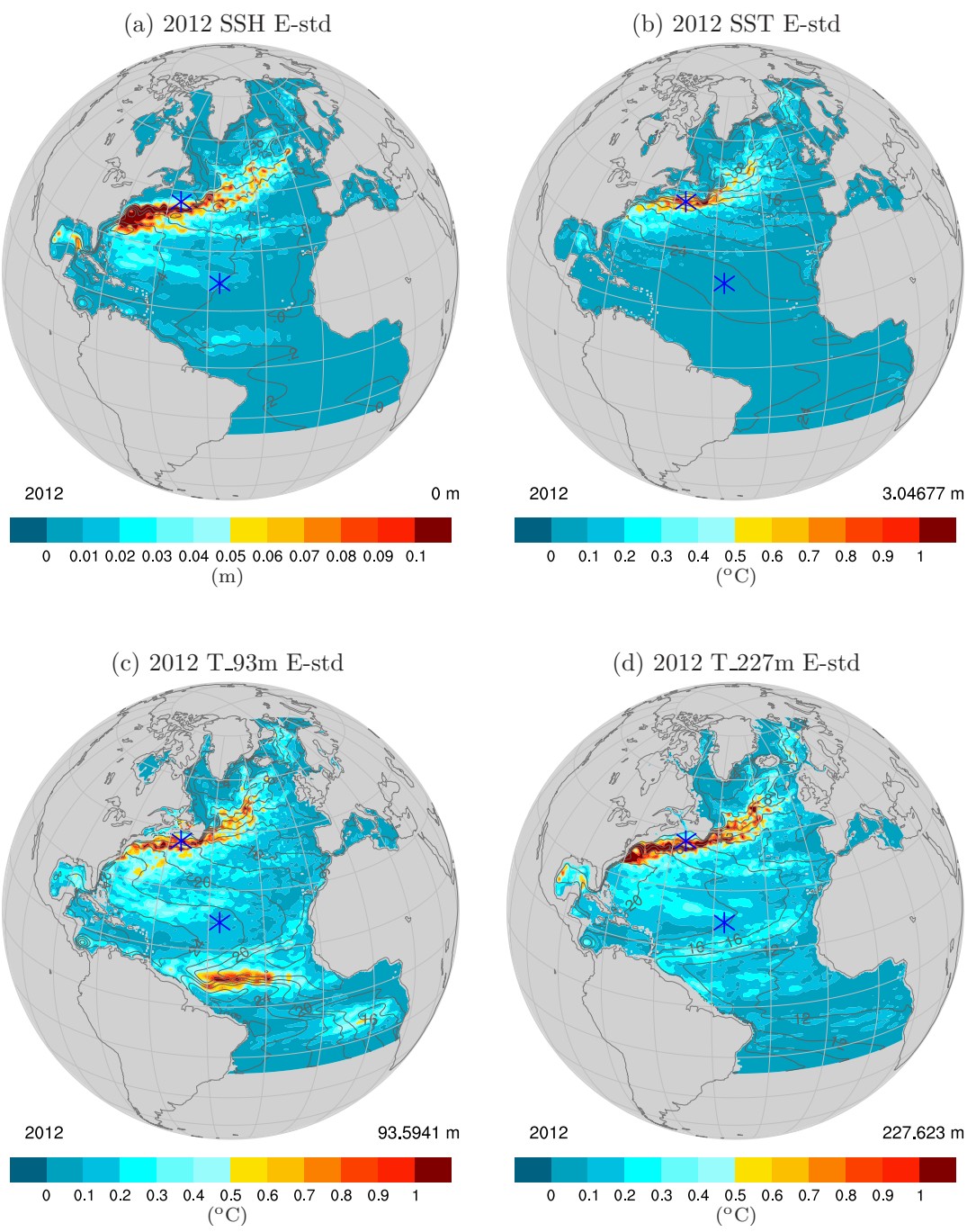

**Figure 5.** E-STD (shading) for year 2012 of the regional ensemble simulation E-NATL025, computed from annual-means of (a) Sea Surface Height (SSH), (b) Sea Surface Temperature, (c,d) Temperature at depths 93 m and 227 m, respectively. The contours show the corresponding E-mean fields. The blue symbols pinpoint the two grid-points at which timeseries are shown in Figure 4.

This is expected from the design of these ensemble simulations: each ensemble member is driven through bulk formulae by the same atmospheric forcing function, but turbulent air-sea heat fluxes somewhat differ among the ensemble because SSTs do so. This approach induces an implicit relaxation of SST toward the same equivalent air temperature (*Barnier et al.*, 1995) within each member, hence an articifial damping of the SST spread. These experiments thus only provide a conservative estimate of the SST intrinsic variability. Note that alternative forcing strategies may alleviate or remove this damping effect: ensemble mean air-sea fluxes may be computed online at each time step and applied identically to all members (see section 3.2). This alternative approach is the subject of ongoing work and will be presented in a dedicated publication.

## 5.4   Magnitudes of forced and intrinsic variability

Panels b and d in Figure 4 show how the E-STD evolves at monthly timescale, and how it compares to various Time-STDs (horizontal straight lines). The Time-STD of E-mean (thick solid yellow line) is a proxy for the amount of the forced variability. It turns out to be dominated by the intrinsic variability (E-STD) at the Gulf Stream grid-point. In less turbulent areas like the subtropical gyre, the intrinsic variability is still about 30-50% of the forced part (Fig. 4.d).

The E-STD can also be compared to the ensemble distribution of the Time-STDs of the $N$ members (see caption of Fig. 4). By construction, the Time-STD of each member is due to both the forced (shared by all members) and the intrinsic (unique to each member) variability. At the Gulf Stream grid-point (Fig. 4.b), these lines all lie above the Time-STD of E-mean, consistent with a high level of E-STD (i.e. intrinsic variability) contributing significantly to the total variability. At the subtropical gyre grid-point, these lines fall much closer to E-mean since little intrinsic variability contributes to the total variability.

## 5.5   Toward probabilistic climate diagnostics

The variability of the Atlantic Meridional Overturning Circulation (AMOC) transport is of major influence on the climate system (e.g. *Buckley and Marshall*, 2016), and is being monitored at 26.5°N since 2004 by the RAPID array (e.g., *Johns et al.*, 2008). These observations are shown at monthly and interannual timescales as an orange line in Figure 6, along with their simulated counterpart from E-ORCA025. They were computed in geopotential coordinates as in Zhang (2010) and *Grégorio et al.* (2015), and are shown after LOESS detrending and after removing the mean seasonal cycle.

The simulated AMOC timeseries are in a good agreement with the observed AMOC variations at both monthly and annual timescales (panels a and c). The total (i.e. combination of forced and intrinsic) AMOC variability is computed as a Time-STD from the observed timeseries and from each ensemble member, and plotted in panels b and d as gray lines. At both time scales, the total AMOC variability simulated by E-ORCA025 lies below the observed variability, consistent with the fact that the model seems to miss a few observed peaks (e.g. 2005, 2009, 2013 on the annual timeseries). Panels b and d also highlight the substantial imprint of chaotic intrinsic variability on this climate-relevant oceanic index at both time scales: at interannual timescale, the AMOC intrinsic variability is weaker than the forced variability, but amounts to about 30% of the latter. A more in-depth investigation of the relative proportion of intrinsic and forced variability in the AMOC and of the variations of the intrinsic contribution with time is currently underway and will be the subject of a dedicated publication.

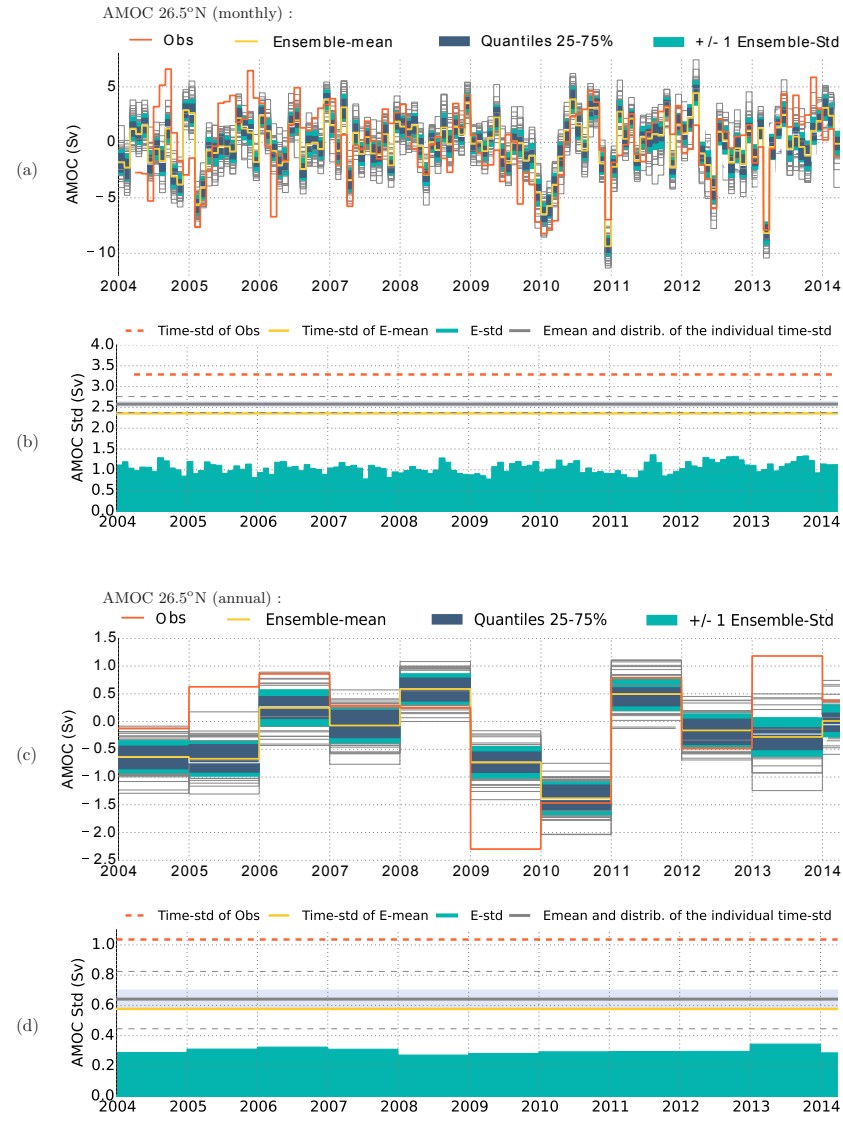

**Figure 6.** Same as Fig.4 but for AMOC anomalies at 26°N in the global ensemble E-ORCA025, from (a,b) monthly- and (c,d) annual-means. In addition, AMOC observational estimates from RAPID at 26°N is shown in orange (see text for details).

# 6    Conclusions

We have presented in this paper the technical implementation of a new, probabilistic version of the NEMO ocean modelling system. Ensemble simulations with $N$ members running simultaneously in a single NEMO executable are made possible through a double MPI parallelization strategy acting both in the spatial and the ensemble dimensions (Fig. 2), and an optimized dimensioning and implementation of the I/O servers (XIOS) on the computing nodes.

The OCCIPUT project was presented here as an example application of these new modelling developments. Its scientific focus is on studying and comparing the intrinsic/chaotic and the atmospherically-forced parts the ocean variability at monthly to multidecadal time scales (e.g. *Penduff et al.*, 2014). For this purpose, we have performed a large ensemble of 50 global ocean/sea-ice hindcasts over the period 1960-2015 at 1/4º resolution, and a reduced-size North Atlantic regional ensemble. These experiments simultaneously simulate the forced and chaotic variabilities, which may then be diagnosed via the ensemble mean and ensemble standard deviation, respectively. The global OCCIPUT ensemble simulation was achieved in a total of 19 million CPU hours on the PRACE French Tier-0 Curie supercomputer, supported by a PRACE grant. It produced about 100 TB of archived outputs.

The members are all driven by the same realistic atmospheric boundary conditions (DFS5.2) through bulk formulae, and represent $N$ independent realisations of the same oceanic hindcast. The ensemble experiments performed here have validated our experimental strategy: a stochastic parametrization was activated for one year to trigger the growth of the ensemble spread (see sections 3.3 and 4.2); the subsequent growth and saturation of the spread is then controlled by the model nonlinearities. Our results also confirm that the spread cascades from short and small (mesoscale) scales to large and long scales. The imprint of intrinsic chaotic variability on various indices turns out to be large, including at large spatial and time scales: the AMOC chaotic variability represents about 30% of the atmospherically-forced variability at interannual time scale. These preliminary results illustrate the importance of this low-frequency oceanic chaos, and advocate for the use of such probabilistic modelling approaches for oceanic simulations driven by a realistic time-varying atmospheric forcing. This approach brings in particular new insights on the imprint of this low-frequency chaos on climate-related oceanic indices, and thus helps anticipate the behavior of the next generation of coupled climate models that will incorporate eddying-ocean components. Ongoing investigations focus on these questions and will be the subject of dedicated papers.

Our probabilistic NEMO version includes several new features. The generic stochastic parameterization, used here on the equation of state to trigger the growth of the ensemble spread, can be applied to other parameters to simulate model or subgrid-scale uncertainties. The MPI communication between members allows the online computation of ensemble statistics (PDFs, variances, covariances, quantiles, etc) across the ensemble members, which may be saved at any frequency, location and for any variable thanks to the flexible XIOS servers.

The size N of the ensemble simulation depends on the objectives of the study, the desired accuracy of ensemble statistics, and the available computing resources. Our choice N=50 allows a good accuracy ($1/\sqrt{50} = \pm 14\%$) for estimating the ensemble means and standard deviations. Moreover, this choice allows the estimation of ensemble deciles (with 5 members per bin) for the detection of possibly bimodal or other non-gaussian features of ensemble PDFs; such behaviors were indeed detected in

simplified ensemble experiments (e.g *Pierini*, 2014) and may appear in ours. Given our preliminary tests with E-NATL025,
N=50 appeared as a satisfactory compromise between our need for a long global 1/4° simultaneous integration, our scientific
objectives, PRACE rules (expected allocation and elapse time, jobs' duration, etc), and CURIE's technical features (processor
performances, memory, communication cost). Our tests also indicate that the convergence of ensemble statistics with N depends
on the variables, metrics and regions under consideration. For all these reasons, N must be chosen adequately for each study.
More generally, this numerical system computes the temporal evolution of the full PDF of the three-dimensional, multivariate
states of the ocean and sea-ice. A very interesting perspective is the online use of the PDF of any state variable or derived
quantity (or other statistics such as ensemble means, variances, covariances, skewnesses, etc) for the computation of the next
time step during the integration. This would allow for instance distinct treatments of the ensemble mean (forced variability)
or the ensemble spread (intrinsic variability) during the integration, e.g. for data assimilation purposes. This NEMO version
can therefore solve the oceanic Fokker-Plack equation, which may open new avenues in term of experimental design for
operational, climate-related, or process-oriented oceanography

## 7   Code availability

The ensemble simulations described in this paper have been performed using a probabilistic ocean modelling system based
on NEMO 3.5. The model code for NEMO 3.5 is available from the NEMO website (www.nemo-ocean.eu). On registering,
individuals can access the code using the open source subversion software (http://subversion.apache.org). The revision number
of the base NEMO code used for this paper is 4521. The probabilistic ocean modelling system is fully available from the
Zenodo website (https://zenodo.org/record/61611) with doi:10.5281/zenodo.61611. The authors warn that this provision of
sources does not imply warranties and support, they decline any responsability for problems, errors, or incorrect usage of
NEMO. Additional information can be found on NEMO website.
The ensemblist features of the model are based on a generic tool implemented in the NEMO parallelization module.
The computer code includes one new FORTRAN routine (mpp_ens_set, see Algorithm 1) which defines the MPI communi-
cators required to perform simultaneous simulations, and to compute online ensemble diagnostics. This routine returns to each
NEMO instance: (i) the MPI communicator that it must use to run the model, and (ii) the index of the ensemble member to be
run. This index can then be used by NEMO to modify: (i) the input filenames (initial condition, forcing, parameters), (ii) the
output filenames (model state, restart file, diagnostics), and (iii) the seed of the random number generator used in the stochastic
parameterizations.
The online computation of ensemble diagnostics requires additional routines, for instance to compute the ensemble mean or
standard deviation of model variables (mpp_ens_ave_std, see Algorithm 2). This routine uses the diagnostic communicators
defined by mpp_ens_set to perform summations over all ensemble members.
As can be seen from these routines, this implementation is generic and can be implemented in any kind of model that is
already parallelized using a domain decomposition method.

---

**Algorithm 1** mpp_ens_set

---

Create world MPI group, including all processors allocated to NEMO ensemble simulation (call to MPI_COMM_GROUP)

**if** (ensemble simulation) **then**

    **for all** (ensemble members $j = 1, \ldots, m$) **do**

        Set the list of processors allocated to member $j$: $r = (j-1) \times n, \ldots, j \times n$

        Create MPI subgroup, including all processors allocated to member $j$ (call to MPI_GROUP_INCL)

        Create MPI communicator, including all processors allocated to member $j$ (call to MPI_COMM_CREATE): $c_{\text{ens}}(j)$

    **end for**

    Get rank of processor in global communicator (call to MPI_COMM_RANK): $r$

    **return** Index of ensemble member to which it belongs: $j = 1 + r/n$

    **return** MPI communicator to be used for this member: $c_{\text{ens}}(j)$

**end if**

**if** (ensemble diagnostic) **then**

    **for all** (subdomains $i = 1, \ldots, n$) **do**

        Set the list of processors allocated to subdomain $i$ (across ensemble members): $r = (i-1) + k \times n,\ k = 1, \ldots, m$

        Create MPI subgroup, including all processors allocated to subdomain $i$ (call to MPI_GROUP_INCL)

        Create MPI communicator, including all processors allocated to subdomain $i$ (call to MPI_COMM_CREATE): $c_{\text{dia}}(i)$

    **end for**

**end if**

---

---

**Algorithm 2** mpp_ens_ave_std

---

**Require:** Array of model variable: $x$

Get diagnostic communicator corresponding to this NEMO instance: $c \leftarrow c_{\text{dia}}(i)$

**if** (ensemble mean) **then**

    Compute sum of $x$ over $c$ (call to MPI_ALLREDUCE, with operation MPI_SUM): $s$

    **return** Mean: $\mu = s/m$

    **if** (ensemble standard deviation) **then**

        Compute anomaly with respect to the mean: $x' \leftarrow x - \mu$

        Compute squared anomaly: $x'^2$

        Compute sum of $x'^2$ over $c$ (call to MPI_ALLREDUCE, with operation MPI_SUM): $s$

        **return** Standard deviation: $\sigma = \sqrt{\frac{s}{m-1}}$

    **end if**

**end if**

---

*Acknowledgements.* This work is mainly a contribution to the OCCIPUT project, which is supported by the Agence Nationale de la Recherche (ANR) through Contract ANR-13-BS06-0007-01. We acknowledge that the results of this research have been achieved using the PRACE Research Infrastructure resource CURIE based in France at TGCC. The support of the TGCC-CCRT hotline from CEA, France to the technical work is gratefully acknowledged. Some of the computations presented in this study were performed at TGCC under allocations granted by GENCI. This work also benefited from many interactions with the DRAKKAR ocean modelling consortium, with the SANGOMA and CHAOCEAN projects. DRAKKAR is the International Coordination Network (GDRI) established between the Centre National de la Recherche Scientifique (CNRS), the National Oceanography Centre in Southampton (NOCS), GEOMAR in Kiel, and IFREMER. SANGOMA is funded by the European Community's Seventh Framework Programme FP7/2007-2013 under grant agreement 283580. CHAOCEAN is funded by the Centre National d'études Spatiales (CNES) through the Ocean Surface Topography Science Team (OST/ST). The authors are grateful for useful comments from three anonymous reviewers; they also thank the NEMO System Team and Yann Meurdesoif for interesting discussions about the development of the probabilistic version of NEMO. LB and SL are supported by ANR. JMB, JMM, PAB, TP and BB are supported by CNRS. MPM and LT are supported by CERFACS, and GS by CNES and Région Midi-Pyrénées.

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
