# Peer review of "Development of a probabilistic ocean modelling system based on NEMO 3.5: application at eddying resolution"

_Geoscientific Model Development, 2016_

## Referee Comment (RC1) · Anonymous Referee #1 · 18 Oct 2016

The authors present the technical implementation of an ensemble version of the NEMO ocean modelling system. Details of the implementation on a parallel architecture are given. Example applications are given consisting of ensemble hindcasts of basin-scale North Atlantic configuration and a global configuration at 1/4 degree. Results are presented from these configurations showing how ensemble statistics generated online can be used to distinguish between forced and intrinsic variability.

Computing power has got to the point where it is becoming feasible to run ensembles of eddying ocean model configurations with a range of possible applications as noted by the authors. I thought this was a very clearly presented description of an ensemble system based on the NEMO model which is likely to be of use to the wider community.

The details of the code availability appear to be complete. The scientific results presented give an enticing prospect of the kind of analysis that will be possible with these new systems.

Minor corrections:

No line for "median" in the legend for Fig 6.

---

## Referee Comment (RC2) · Anonymous Referee #2 · 13 Nov 2016

The paper Âń Development of a probabilistic ocean modelling system with NEMO at eddying resolution" presents a really innovative and useful method to produce ocean ensemble and associated diagnostics and statistics. The subject and the way it is presented are perfectly in the scope of GMD and especially for the NEMO special issue. I have only few recommendations and questions that authors can take into account in the final version of the paper. 1) The online diagnostics is one of the most useful developments proposed in the paper but authors don't provide any information on the computational cost of these online diagnostics. As these diagnostics need several global mpi communications, the cost should be important. Could you provide this cost at least for an example of this statistic? 2) Authors suggest that the ensemble online

[Figure]

method could be useful for relaxation of the ensemble mean toward a climatology for example. Could you explain more precisely the way this could be done, is there already work and references about such method? It is not obvious that it will work properly. Is there a way to keep a good spread of the ensemble ? 3) Could you explain why do you use the NATL experiment for the gulf stream study and the ORCA one for the MOC? 4) Could you provide more information of the restoring which is done in the simulation? Is there a sea surface salinity restoring, a sea surface temperature restoring? 5) As you use bulk formulae to compute your atmospheric fluxes and to constrain your model, it is not true that you have strictly the same atmospheric forcing in all the members. Could you provide quantified informations of the variance of the atmospheric fluxes in the experiment? It will be useful to know if this variability is negligible or not. 6) There is no discussion about impact of the number of members in the study, as you have 50 members in your global simulation it will be interesting to know how each member gives information and if the ensemble spread converges? This point could at least discussed in the perspectives.

Figure 3 : Keeping the same color or symbol code between fig 3a and 3b could be more clear for reader Figure 6 a) there is no legend line for the Median.
* * *

---

## Referee Comment (RC3) · Anonymous Referee #3 · 15 Nov 2016

The paper "Development of a probabilistic ocean modelling system based on NEMO 3.5: application at eddying resolution" by Bessières et al. presents an ensemble-simulation strategy in the NEMO ocean model. Results from ensemble ocean hindcasts are presented to illustrate the utility of the software. The material is a good fit for this journal and will no doubt be of wide interest. I have several 'major' criticisms, but they relate only to the way in which the broader context is discussed, and not to the specific results or to the software.

**Major**

- Model uncertainties can be modeled in a huge variety of ways. The discussion surrounding equation (3) tacitly suggests that the only, best, or most common way to model model uncertainty is by adding Gaussian white noise to the time tendency equation. Many other options exist; for example using non-Gaussian white noise, colored noise, or forcing with the increments of a Levy process instead of a Weiner process, to name just a few (other options exist even beyond adding terms to the equations, for example when a parameter is unknown but should not be time-dependent). The stochastic parameterizations used in NEMO in this paper (as far as I can tell) are not in the form of equation (3) because the noise terms are generated by time-correlated processes (either AR-1 or piecewise-continuous in time). (I grant that for an AR-1 noise process you can expand the definition of $\mathbf{x}$ such that the whole system is in the form (3).) In general, a stochastic model has a master equation that models the evolution of the probability density, but this master equation need not be a Fokker-Planck equation. Tying the discussion to an SDE forced by Weiner-process increments is overly restrictive and the ensemble framework described in this paper is just as useful in a broader context as it is in the restricted context.

- In a few places it is claimed that (i) an ensemble simulation provides a solution to the Fokker-Planck equation, or equivalently that (ii) the ensemble provides an approximation to the probability density function $p$ on the model state $\mathbf{x}$ at a specific time $t$. I suppose that there is a sense in which this is true, but with any realistic ensemble size the approximation to $p$ is horrendously bad. I suggest that a more accurate phrasing would be to say that an ensemble provides a set of independent, identically-distributed (iid) draws/samples/realizations from the distribution, and that these draws/samples/realizations can be used to compute a Monte-Carlo approximation to any statistic of interest, like the mean, covariance, or even the pdf itself. The accuracy of the Monte Carlo approximation depends on the number of samples (ensemble size), the distribution from which they're drawn,
and the quantity of interest. For example, the ensemble mean can probably be estimated fairly well, but the full pdf $p(\mathbf{x}, t)$ almost certaintly cannot.

Furthermore, the Fokker-Planck equation (or, more generally, the master equation associated with the stochastic model) only provides a pdf on the model state at a particular time, which is again overly-restrictive. The pdf at a particular time (even if you knew what it was) cannot be used to estimate things like temporal correlations, but such things *can* be estimated (using Monte-Carlo) from the ensemble, because the full time history of each ensemble member is a draw/sample/realization from the more general joint distribution of the model variables at different times.

In summary, I think it would be better to describe the ensemble system as providing a set of iid samples that can be used to compute Monte-Carlo approximations to any statistical quantity of interest.

- I think the term 'equiprobable,' frequently used to describe the ensemble members, is at best misleading. It suggests that the ensemble members are equally probable. This is usually not true. Rather, the ensemble members are independent and identically distributed (iid), and within a Monte-Carlo context they all have equal weight (perhaps the equal weights are what is meant here by equiprobable). For example, the only way to draw 'equiprobable' samples from a scalar normal distribution is for the samples to all be located an equal distance from the mean, whereas any iid set of samples – some highly improbable, others highly probable – can be used in a Monte-Carlo approximation with equal weights.

---

## Author Comment (AC1) · 9 Dec 2016

In the following, the reviewers' questions and comments are shown in bold-italic type, our answers appear in standard type and the modified text of the manuscript is given in italic type.

**Response to REVIEWER #1:**

*1) No line for "median" in the legend for Fig 6.*

Fig. 6: Thank you for noting this. In fact the ensemble median was plotted as a thin white line, which is why it is not seen in the caption. Most of the time, the median is anyway hidden below the line of the ensemble mean (thick yellow) in the figure, since

[Figure]

the ensemble PDF is quasi-gaussian. So to avoid confusion we have now removed 'median' from the figure and caption.

NB: We wish to thank the reviewers for their interest in our paper, for their constructive comments and suggestions that lead to useful improvements in the manuscript.

---

## Author Comment (AC2) · 9 Dec 2016

In the following, the reviewers' questions and comments are shown in bold-italic type, our answers appear in standard type and the modified text of the manuscript is given in italic type.

**Response to REVIEWER #2:**

*1) The online diagnostics is one of the most useful developments proposed in the paper but authors don't provide any information on the computational cost of these online diagnostics. As these diagnostics need several global mpi communications, the cost should be important. Could you provide this cost at least*

[Figure]

***for an example of this statistic?***

We have computed online the ensemble mean and variance of four 2D fields (i.e mixed layer depth, sea-surface temperature, sea-surface zonal and meridional velocities) at both daily and monthly frequencies during the whole OCCIPUT global run; the same diagnostics were also computed at hourly frequency for 6 specific months. Our tests with and without this small set of online ensemble diagnostics showed that they did not induce any noticeable increase of the cost, which remained undistinguishable from the slight, random run-to-run variations of CPU costs.

More generally, the cost of these diagnostics depends at first order on their call frequency and on the dimension of the treated arrays: their cost may indeed become noticeable or even substantial if online ensemble diagnostics were applied on all 3D fields at all time steps. We did not perform such an extreme test and thus we cannot evaluate its computational cost. We have added the following paragraph in section 4.3 to address and summarize these points:

*"The cost of online ensemble diagnostics depends on the call frequency, number and size of the concerned fields, on the architecture of the machine and the performance of communications. Our online ensemble diagnostics concerned a few two-dimensional fields at hourly to monthly frequencies, and had a negligeable cost. "*

***2) Authors suggest that the ensemble online method could be useful for relaxation of the ensemble mean toward a climatology for example. Could you explain more precisely the way this could be done, is there already work and references about such method? It is not obvious that it will work properly. Is there a way to keep a good spread of the ensemble ?***

The comment you refer to in section 3.2 is an example that illustrates how the online ensemble diagnostics may be used. By "relaxation of the ensemble mean toward a climatology" we mean computing a correction term based on the simulated ensemble mean, and then applying this term identically to all members. By construction, each

member would be corrected by the same amount: this would not directly affect the ensemble spread but only "translate" the entire ensemble distribution toward the climatological value. We are not aware of any reference about such an approach, but implementing it would be straightforward. Note that we discuss a variant of this method in our response to your question 5) about the surface fluxes. We have tried to clarify this example in section 3.2. Therefore, the following paragraph:

*"This may be useful for certain applications: a simple example would be for example the relaxation (nudging) of the model simulation towards some climatological data. In this case, indeed it could be much better to relax the ensemble mean than the individual ensemble members, to avoid damping the intrinsic variability of the system by the relaxation."*

has been replace by:

*For instance, it may be interesting to relax the modeled forced variability towards reference (e.g. reanalyzed or climatological) fields, with no explicit damping of the intrinsic variability: the nudging term would involve the current ensemble mean and be applied identically to all members at the next time step, resulting in a simple "translation" of the entire ensemble distribution toward the reference field.*

**3) Could you explain why do you use the NATL experiment for the gulf stream study and the ORCA one for the MOC?**

The objective of the paper is mainly to present new, generic model developments implemented in NEMO. The NATL and ORCA experiments are only given here as examples, to validate the ensemble modeling system in both its regional and global configurations, and to illustrate different types of results it can provide on various ocean quantities and scales (monthly temperatures, annual MOC, etc). Given the high computational cost of such ensemble experiments, the choice of using regional instead global configurations may be judicious in some cases, depending on the scientific questions that are addressed. Some studies require the global configuration anyway; for example, the

global ensemble simulation shows that part of the interannual intrinsic variability (ensemble spread) of the AMOC is generated in the South Atlantic, and is thus missing in the regional ensemble experiment. We are currently working on a publication dedicated to this subject (Leroux et al, 2017, in prep for J. of Climate) as also mentioned in the last paragraph of section 5.

*4) Could you provide more information of the restoring which is done in the simulation? Is there a sea surface salinity restoring, a sea surface temperature restoring?*

As in most oceanic hindcasts, there is a SSS restoring within each member that corresponds to a 300 days timescale over 50m (166.67 mm/day piston velocity; see Griffies et al, Ocean Modelling 2009). This information is now given in Table 1. There is no explicit SST relaxation; please see our answer to the next question regarding the implicit SST relaxation due to the use of bulk formulae.

*5) As you use bulk formulae to compute your atmospheric fluxes and to constrain your model, it is not true that you have strictly the same atmospheric forcing in all the members. Could you provide quantified informations of the variance of the atmospheric fluxes in the experiment? It will be useful to know if this variability is negligible or not.*

Indeed, the atmospheric fluxes computed through bulk formulae somewhat differ among the ensemble because SSTs do so. However all members "see" the exact same atmospheric evolution, as we wrote in section 4.2 (*"forced by the exact same atmospheric conditions"*). We have clarified this point in the last paragraph of section 5.3, which now reads:

*"This is expected from the design of these ensemble simulations: each ensemble member is driven through bulk formulae by the same atmospheric forcing function, but turbulent air-sea heat fluxes somewhat differ among the ensemble because SSTs do so. This approach induces an implicit relaxation of SST toward the same equivalent air*

*temperature (Barnier et al, 1995) within each member, hence an articifial damping of the SST spread. These experiments thus only provide a conservative estimate of the SST intrinsic variability. Note that alternative forcing strategies may alleviate or remove this damping effect: ensemble mean air-sea fluxes may be computed online at each time step and applied identically to all members (see section 3.2). This alternative approach is the subject of ongoing work and will be presented in a dedicated publication."*

**6) There is no discussion about impact of the number of members in the study, as you have 50 members in your global simulation it will be interesting to know how each member gives information and if the ensemble spread converges? This point could at least discussed in the perspectives.**

Our paper mostly aims to present the probabilistic version of NEMO; choosing an appropriate number of ensemble members is an important concern for users, depending on their specific applications. This question is now shortly discussed in the conclusion, as follows;

*The size N of the ensemble simulation depends on the objectives of the study, the desired accuracy of ensemble statistics, and the available computing resources. Our choice N=50 allows a good accuracy ($1/\sqrt{50} = \pm14\%$) for estimating the ensemble means and standard deviations. Moreover, this choice allows the estimation of ensemble deciles (with 5 members per bin) for the detection of possibly bimodal or other non-gaussian features of ensemble PDFs; such behaviors were indeed detected in simplified ensemble experiments (e.g. Pierini, 2014) and may appear in ours. Given our preliminary tests with E-NATL025, N=50 appeared as a satisfactory compromise between our need for a long global 1/4° simultaneous integration, our scientific objectives, PRACE rules (expected allocation and elapse time, jobs' duration, etc), and CURIE's technical features (processor performances, memory, communication cost). Our tests also indicate that the convergence of ensemble statistics with N depends on the variables, metrics and regions under consideration. For all these reasons, N must be chosen adequately for each study.*

**Other comments on figures**

*Fig. 3: Keeping the same color or symbol code between fig 3a and 3b could be more clear for reader*

As suggested, the same color code has been kept between fig 3a and 3b.

*Fig. 6 : There is no legend line for the Median*

See our answer to reviewer 1 question 1.

NB: We wish to thank the reviewers for their interest in our paper, for their constructive comments and suggestions that lead to useful improvements in the manuscript.

––––––––––––––––––––––

---

## Author Comment (AC3) · 9 Dec 2016

In the following, the reviewers' questions and comments are shown in bold-italic type, our answers appear in standard type and the modified text of the manuscript is given in italic type.

**Response to REVIEWER #3:**

**1) Please refer to interactive discussion for referee#3 comment#1**

We agree that the mathematical background given in section 2 does not provide the most general mathematical framework, and does not encompass all possible ways of simulating model uncertainties. However, on the one hand, it is general enough to

include everything that it is possible to do with our specific implementation. Our implementation indeed only introduces autoregressive processes (time-correlated and/or space-correlated), possibly followed by a nonlinear transformation to make them non-Gaussian, and all this is included in the given mathematical framework (by changing the definition of x and M as granted by the reviewer). And, on the other hand, using a more general mathematical framework would make this section more difficult to follow for non-mathematicians.

To answer the reviewer comment, we have thus decided to keep a simple mathematical framework, but we have added a word of caution explaining that this is only one possible approach, and an explanation that the ensemble method could also be used if another type of stochastic parameterization had been implemented.

(i) Before Eq. 3, the following sentence has been added: *"One possibility is for instance to modify Eq. 1 as follows:"*

(ii) After Eq. 3, the following sentence has been added: *"Eq. (3) does not include all possible ways of introducing a stochastic parameterization in a dynamical model, but it is sufficient to include the implementation that is described in this paper (in particular, to include the use of space-correlated or time-correlated autoregressive processes by expanding the definition of $x$ in Eq. (3))."*

(iii) The last sentence of the 4th paragraph of section 2 (before the final summary paragraph) has been rewritten: *"This Monte Carlo approach is very general and be also applied to any kind of stochastic parameterization (not only the particular case described by Eq. 3). It was first applied (...)."*

**2) Please refer to interactive discussion for referee#3 comment#2**

Yes, we agree that the ensemble usually provides only a rough approximation of the probability distribution, and that the ensemble can give estimates of quantities (like time correlations) that are not contained in the marginal pdfs (for each time t) provided by
the Fokker-Planck equation.

To clarify this, the following changes have been made in the text of the paper:

(i) the last sentence of section 2 has been replaced by *"in this paper, this problem is solved using an ensemble simulation, which provides identically-distributed realizations from the probability distribution, and thus a way to compute any statistic of interest."*

(ii) in the 3rd paragraph, *"a solution to Eq. (2) or (4)"* has been replaced by *"an approximate description of the probability distribution"*.

(iii) in section 3.3, the first sentence has been replaced by: *"Ensemble simulation (...) parameterization (as introduced in Eq. 3)."*

**3) Please refer to interactive discussion for referee#3 comment#3**

Yes, we agree with the reviewer, the term *"equiprobable"* was not used correctly. It has been replaced by *"independent and identically distributed"* as suggested by the reviewer, or just suppressed where it was not useful.

NB: We thank the reviewer for his/her interesting comments about the mathematical formulation in our paper. They helped us improving the generality and accuracy of our statements. We did our best to take them into account.
* * *